# The Effect of TiN, TiAlN, TiCN Thin Films Obtained by Reactive Magnetron Sputtering Method on the Wear Behavior of Ti6Al4V Alloy: A Comparative Study

**Şengül Danışman** [1,*], **Durmuş Odabaş** [1] and **Muharrem Teber** [2]

[1] Department of Mechanical Engineering, Erciyes University, 38039 Kayseri, Turkey
[2] HES Hacılar Electricity Industry, Trade Inc., 38210 Kayseri, Turkey
[*] Correspondence: sdanisman@erciyes.edu.tr

**Abstract:** Titanium-based alloys have gained worldwide application over the past century. However, their low wear resistance remains an unresolved challenge for the Ti6Al4V alloy, which has significant industrial use. Therefore, it is an integral part in material selection and surface treatment in friction-wear applications. Tribological properties are not only material parameters but also system parameters where test conditions are essential. Hardness, roughness and contact conditions of coatings, which are especially important in surface treatments, affect wear modes. In this study, titanium nitride, titanium aluminum nitride and titanium carbon nitride coatings were obtained by unbalanced reactive magnetron sputtering to improve the weak tribological properties of Ti6Al4V alloy. The wear behavior was studied at room temperature in dry conditions. Wear tests were conducted under different loads and different sliding rates, which were followed by comparative analyses of their wear resistances. While the coated samples showed higher wear resistance than the uncoated Ti6Al4V alloy, the lowest wear track width was observed in TiN coating. Depending on the surface properties of the coatings, adhesion, abrasion and oxidation wear mechanisms were observed. It was concluded that a TiN coating could be a material of choice for applications where triple abrasive wear is dominant.

**Keywords:** biomedical; Ti6Al4V alloy; TiN; TiAlN; TiCN coatings; wear

## 1. Introduction

Although the Ti6Al4V alloy was developed in 1954, it is a material that maintains its importance in many industrial sectors today [1–3]. Low density, high strength to weight ratio, machinability, and oxidation resistance are essential features of this alloy [4,5]. A wide range of industries benefit from these features such as transportation, aviation, and space industries. At the same time, the Ti6Al4V alloy is one of the prominent materials in the chemistry and biomedical fields and meets essential requirements, such as fatigue and corrosion resistance [4,6–10]. Although the Ti6Al4V alloy has a 75–85% consumption among titanium alloys, it has some limiting disadvantages, such as a high friction coefficient and poor tribological properties [1–4]. Improving surface properties and wear resistance in the fields where Ti6Al4V alloy is used is critical [4,11–13]. Studies carried out for this purpose can be grouped under two headings: (i) making materials resistant to abrasion or (ii) applying surface treatments to increase wear resistance.

One of the latest applications to make the Ti6Al4V alloy wear resistant is to deform it by high pressure torsion (HPT). While the microhardness of the nanostructured alloy obtained by this process increases by about 41%, its tribological properties improve under dry conditions, and the coefficient of friction and the specific wear rates decrease [14]. As the number of revolutions increases in the HPT process, the average grain size obtained in the Ti6Al4V alloy decreases. It is stated in the literature that grain sizes between 30 and 300 nm can be obtained with up to ten or more revolutions [14].

Another approach is the acquisition of wear resistant surfaces on the base material without compromising the properties of the host material. Thin-film hard nitride coatings can be given as an example which is industrially active for surface engineering of Ti6Al4V alloy [15]. High hardness is attributed to the micro and nanostructures obtained in thin film nitride coatings [16,17]. At the same time, thin film nitride coatings of titanium alloys are used as barriers to prevent diffusion [18]. The most effective barriers to diffusion in medical applications are ultrafine grain coatings [18,19]. The appropriate coating method is vital to obtain the desired fine-grained, columnar structure model [19,20].

There are many coating methods to increase the wear resistance of Ti6Al4V alloy. Chemical vapor deposition (CVD) and physical vapor deposition (PVD) are the leading methods. CVD has a high deposition rate and good coating uniformity. However, it requires higher deposition temperatures and higher gas pressures than PVD. PVD is gaining importance due to its low deposition temperature, controllable coating composition, and being a waste-free method in terms of the environment [21,22]. Other coating methods are spraying, sol-gel dipping and electrophoretic deposition (EPD) [23–29]. Titanium nitride-based coatings are deposited by magnetron sputtering, cathodic arc and pulsed laser deposition techniques, among the PVD techniques [15]. The cathodic arc method provides a low voltage, high current plasma discharge, but the surfaces are rough due to droplet formation [30]. Compared with the cathodic arc method, the D.C. reactive magnetron sputtering method can smoothly cover more extensive areas at lower temperatures [21,31]. This method provides the desired fine-grained and columnar structures depending on the coating process parameters [18,20]. Especially in Thornton's structure region model [20], coarse-grained, fine-grained and columnar coating microstructures can be obtained by controlling the ratio of argon pressure and substrate temperature to the melting temperature of the coating material. In industrial applications, the structure pattern of coatings closely affects the wear resistance [15].

Coating materials such as titanium nitride (TiN), titanium carbo-nitride (TiCN), titanium carbide (TiC), chromium nitride (CrN) and dimond-like carbon (DLC) are widely used to improve the surface quality [21,32–36]. Titanium aluminum nitride (TiAlN) coatings obtained by adding Al to form ternary thin films are particularly attractive. It was developed as an alternative to TiN due to its higher oxidation and corrosion resistance as well as high hardness. Nitride-based coatings are particularly popular for having applications in various demanding fields. The current study aims to quantitatively investigate the surface improving capabilities of these coatings.

In this study, it is attempted to improve the weak wear resistance of the Ti6Al4V alloy by acquiring TiN, TiAlN, and TiCN coatings using the closed field unbalanced magnetron sputtering method. Microstructural analysis of the surfaces was performed using scanning electron microscope (SEM), energy dispersive X-ray (EDX), atomic force microscope (AFM) and X-ray diffraction (XRD) for the detailed characterization of the hard coatings. Wear behavior of the coated materials was studied under dry conditions at various parameters such as different loads and sliding rates. Wear losses and the widths of the wear tracks were determined after conducting wear tests. Wear tests were performed by the pin-on-ring method. The wear behavior of various nitride coatings was not only compared with each other but also to the uncoated reference materials.

## 2. Materials and Methods

### 2.1. Sample Preparation and Coating Method

This study used a dual-target unbalanced magnetic sputtering system to realize hard and wear resistant coatings (TiN, TiAlN and TiCN) on the Ti6Al4V alloy. The exact composition of the host Ti6Al4V is given in Table 1. Before coating, plasma nitriding process was applied to the host Ti6Al4V samples ($20 \times 65 \times 3$ mm) to increase their surface hardness. The process was carried out at 650 °C for 1 h in the presence of 25% Ar + 75% $N_2$. This was followed by the grinding and polishing of the samples' surfaces. Later, both silicon wafers and Ti6Al4V samples were cleaned in a "Bandelin Sonorex RK 255-H" brand ultrasonic

bath. Cleaning is essential for the adhesion of coatings to the host material. Cleaning continued for 15 min in an environment containing acetone and alcohol. Samples were later rinsed with distilled water, dried and placed in a vacuum chamber.

**Table 1.** Chemical compositions of Ti6Al4V samples (wt.%).

| Element (%) | N | Fe | Al | V | C | H + O | Ti |
|---|---|---|---|---|---|---|---|
| Ti6Al4V | <0.05 | 0.25 | 6.11 | 3.78 | <0.08 | <0.3 | Balance |

The sputtering method is kinetically controlled. The cathode, called the target, is bombarded by the energetic ions produced in the glow discharge plasma, thus removing many target particles. The magnetic field used in the magnetron sputtering method captures high-energy electrons in front of the target, thereby increasing the target ionization rate [21]. The two-target unbalanced magnetron sputtering system used in this study increases the plasma density, and better coating quality and deposition rates are obtained [37,38]. The vacuum chamber and magnetron sputtering coating system [37,38] are shown schematically in Figure 1.

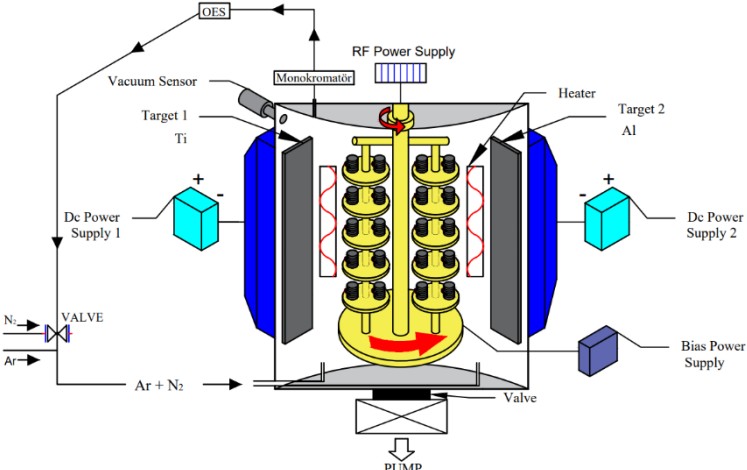

**Figure 1.** Schematic view of the vacuum chamber and magnetron sputtering coating system.

Coatings were deposited on the substrate materials in this system by controlling the target power, bias voltage, and gas pressures [37,38]. The coating process was performed with the following steps:

Before starting the coating, sputter cleaning was performed to remove contamination from the surfaces of the substrate materials placed in the rotary holders in the vacuum chamber. Sputter cleaning was applied to the test samples at 500 W RF power and 5 mtorr argon pressure for 25 min. Then, target cleaning was performed for ~5 min at 4500 W and 5 mtorr argon pressure.

Coating processes were carried out in two stages. First, a titanium interlayer was coated on the substrates to provide better adhesion to the surface of the substrate material. This process was applied for 10 min at 2 mtorr argon pressure using 4000 W target power. Then, the reactive coating processes were carried out at 5000 W target power, −80 V bias voltage, and three hours of coating time. TiN and TiAlN coatings were carried out in an argon +$N_2$ atmosphere. Nitrogen gas was controlled with an optical emission spectrometer to prevent target poisoning. The TiAlN coating was deposited on the Ti interlayer. TiCN coatings were deposited by using a pure titanium target (99.9%) onto the substrate in Ar/$N_2$ and $C_2H_2$ gas atmospheres. TiCN coating was acquired in the order of Ti interlayer + TiN coating + TiCN coating. After the coating process, uncoated Ti6Al4V alloy, TiN, TiAlN and TiCN coatings were subjected to wear tests in four groups under the same conditions.

## 2.2. Wear Tests

The wear tests in this study were carried out in a dry environment at room temperature. The samples were examined in four groups: uncoated Ti6Al4V alloy, TiN coated Ti alloy, TiAlN coated Ti alloy, and TiCN coated Ti alloy. For the wear tests, a Plint brand TE 53 SLIM multi-purpose friction and abrasion test device was used as shown in Figure 2a.

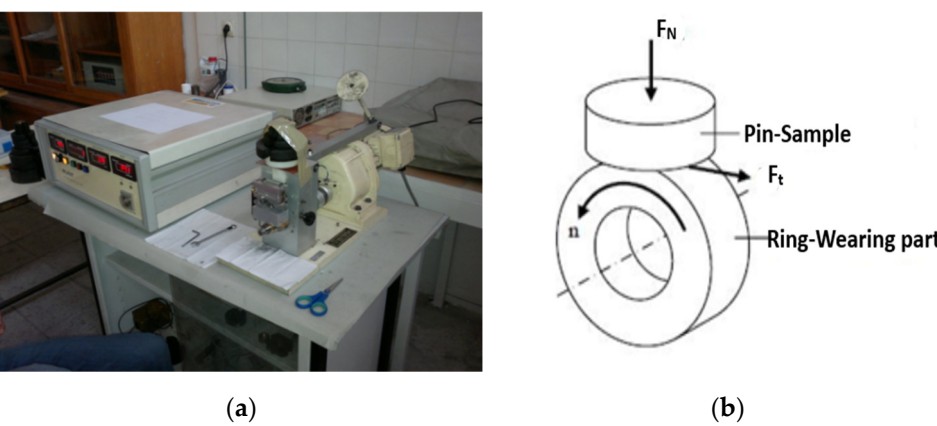

(a)                                                                    (b)

**Figure 2.** (**a**) The pin-on-ring test device used in the wear test, (**b**) Pin, ring contact geometry and forces.

The wear behavior of the four sample groups was investigated using a wear tester adapted to pin-on-ring system (ASTM G-77), as shown in Figure 2b. Three different loads (5, 15, and 30 N) and two different sliding rates (0.5 m/s and 0.9 m/s) were chosen as wear test parameters. The abrasive disc material used in wear tests is DIN 100Cr6 (bearing outer ring) and has a diameter of 26 mm and a height of 5 mm. In addition, it has a hardness of 61 RC and a surface roughness of Ra 0.12 μm. The chemical composition of the selected abrasive disc is as follows: C 0.99%, Mn 0.38%, Cr 1.42%, Mo 0.02%, and the remainder Fe. The experiments were carried out using a fixed wear time of 15 min.

## 3. Experimental Results and Discussion

### 3.1. Coating Characterization

This study performed the thickness, hardness, roughness, SEM, EDX, AFM, and XRD analyses to characterize the coatings.

The thickness of TiN, TiAlN, and TiCN thin films was measured with the CSEM Calotest device which ranged from 1.75 μm to 2 μm. At the same time, TiN, TiAlN, and TiCN thin films were obtained on silicon wafers, and the coating thicknesses were displayed by examining the cross-sections of silicon wafers in SEM. Cross-sectional images of the samples were recorded with a scanning electron microscope (SEM-LEO 440 Computer Controlled Digital model, Electron Microscopy Ltd., Cambridge, MA, USA). The SEM image given in Figure 3 shows that the TiCN coating thickness is homogeneous and uniform. EDX analysis taken in the coating region gives the elemental composition of the TiCN coating. It also turns out that the coating throughout the thickness exhibits a dense, columnar crystal structure, as noted in the literature [39]. The EDX analysis reveals the presence of a Ti interlayer + compound coating layer. The Ti interlayer applied to the base material increases the bond strength and provides a simple approach to control residual stresses due to the stress-buffering effect [40,41].

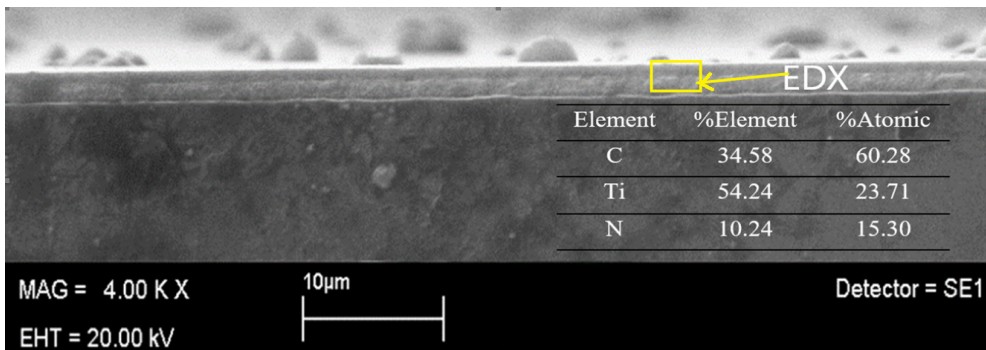

**Figure 3.** Scanning electron microscope image and energy dispersive X-ray analyses of the TiCN coating thickness.

The hardness measurement of the samples prepared from Ti6Al4V alloy was determined at two stages, before and after coating. In addition, hardness measurements were made before and after plasma nitriding to see the effect of plasma nitriding on hardness before the application of coating. The hardness of the samples before and after plasma nitriding was found as 384 $HV_{0.1}$ (Vickers hardness) and 592 $HV_{0.1}$, respectively, based on the Vickers method. The average of three values was used in the measurements made using the Struers brand microhardness device.

This study measured the hardness of TiN, TiAlN, and TiCN thin films obtained after the coating process using a CSEM brand nano hardness tester. The basic principle to avoid the influence of the substrate material in the hardness measurements of the coatings is that the depth of penetration of the indenter should not exceed [42] 10% of the coating thickness. According to the Oliver-Pharr method, hardness measurements were made using a 15 mN load [42]. The suitability of the load selection was checked with the load-penetration depth curves obtained from the hardness measurements made with a square-based diamond Vickers tip ($\alpha$ = 136°). The hardness values of the coated samples were the average of 5 measurements and were found to be approximately 3500 HV for TiN [43], 1960 HV for TiAlN and 1050 HV for TiCN.

Roughness measurements were made with the Mitutoyo Surftest roughness device. First, surface roughness values were taken before coating the samples prepared from Ti6Al4V alloy. Then, the roughness measurements of the obtained TiN, TiAlN, and TiCN thin films were compared. The average surface roughness values of the samples are shown in Table 2.

**Table 2.** Average surface roughness Ra (µm) values for Ti6Al4V alloy and coatings.

| | Ra (µm) | | | | |
|---|---|---|---|---|---|
| **Test Sample Number** | **1** | **2** | **3** | **4** | **Average Values** |
| Ti6Al4V | 0.31 | 0.54 | 0.38 | 0.52 | 0.44 |
| TiN | 0.19 | 0.66 | 0.47 | 0.56 | 0.47 |
| TiCN | 0.18 | 0.26 | 0.21 | 0.17 | 0.21 |
| TiAlN | 0.55 | 0.51 | 0.24 | 0.30 | 0.4 |

Surface roughness is vital in determining the mechanical and wear behavior of coatings. While coating materials affect the roughness value of the exposed surfaces, a low surface roughness is preferred because it improves wear performance [44,45]. Table 2 shows the surface roughness values of the various coatings in which the highest roughness value is witnessed for the TiN coating.

The surface roughness of Ti alloy, TiAlN, TiCN and TiN are shown in Figure 4 as examined through AFM. The results obtained from the AFM agree with the roughness measurements acquired with the profilometer. AFM images revealed that the least surface roughness was measured for TiCN.

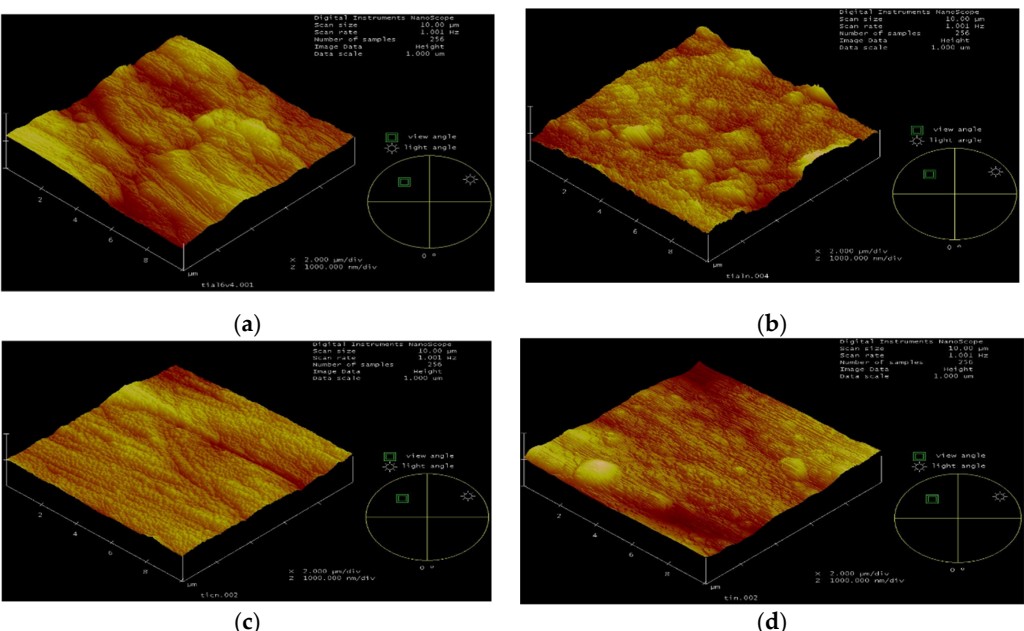

**Figure 4.** Atomic force microscope images; (**a**) Ti6Al4V, (**b**) TiAlN, (**c**) TiCN, (**d**) TiN.

AFM images show that a dense and crack-free surface structure is obtained after the coating. The surface contains buds which might have formed due to formation of thinner and columnar structures. The cross-sectional scanning electron microscope image of the TiN coating is shown in Figure 5a, and the qualitative analysis of EDX obtained in the marked region is shown in Figure 5b. According to this analysis, 90.75% Ti and 9.25% N were determined. Similarly, EDX analysis of the Ti6Al4V alloy shows 89.46% Ti, 6.02% Al and 4.51% V. EDX analysis of the TiCN coated Ti alloy has values of 31.48% C, 61.06% Ti and 7.46% N. EDX analysis obtained for TiAlN coated samples show 39.87% Al, 33.86% Ti and 25.82% N.

X-ray diffraction (XRD) analysis was performed using the BRUKER AXS D8 Advance Model (Cu Tube, Wavelength 1.5406 Angstrom, 40 kV, Bruker AXS GmbH, Karlsruhe, Germany) device. Diffraction peak angles and intensities of Ti alloy and coatings were determined. Figure 6 shows the XRD patterns of the uncoated Ti alloy and TiN, TiAlN, and TiCN films. The comparative XRD results reveal the preferential orientation of the coatings depending on the 2θ angle value. Each diffraction profile defines its unique crystal structure. When the XRD pattern of Ti6Al4V alloy is examined, it is revealed that it is displayed with many phases. The plane (101) at 40.23° and the plane (002) at 38.23° are the highest peaks showing strong orientation. In addition, there are reflection planes such as (100), (102), (110), (103), (112) and (201), which are determined by observing the orientation peak and noticed to be weaker than other peaks.

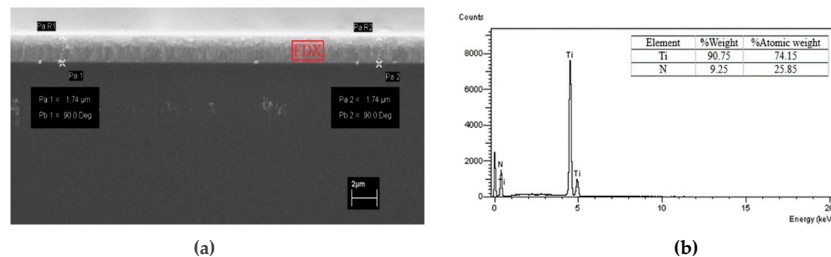

| Element | %Weight | %Atomic weight |
|---------|---------|----------------|
| Ti | 90.75 | 74.15 |
| N | 9.25 | 25.85 |

**Figure 5.** (**a**) Cross-sectional scanning electron microscope image of the TiN coating, (**b**) Energy dispersive X-ray analysis.

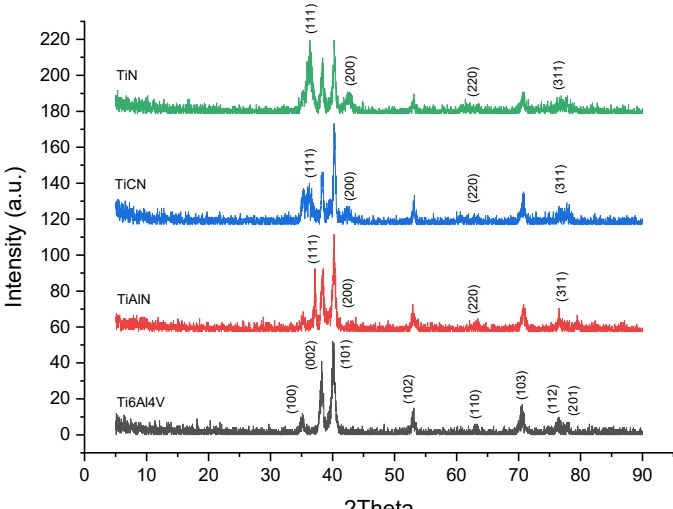

**Figure 6.** X-ray diffraction analysis patterns of the uncoated Ti alloy and TiN, TiAlN, TiCN films deposited on Ti6Al4V.

It is known that the Ti-6Al-4V alloy has an α + β phase structure. In the alloy, vanadium promotes the β-BCC phase at room temperature, a beta stabilizer element [46]. However, no β-BCC lattice structure was observed in the X-ray diffraction pattern of the Ti6Al4V sample. The strong intensity peaks have a tightly packed hexagonally alpha phase [46]. The hexagonal α phase is defined for the Ti6Al4V(JCPDF#44-1294) alloy. When the XRD patterns of the coatings were compared, significant intensity orientations were observed, especially in the (111) and (200) planes.

This study revealed that the proximity of the diffraction peaks in the XRD analysis of the coatings indicates a crystal structure similarity between them. XRD analysis of TiAlN, TiN and TiCN coatings shows that the lattice structure of the coatings is completely cubic in contrast to the alloy [47]. In the XRD analysis of TiN(JCPDF#38-1420) coating, the 2θ 36.28° (111) aspect and 42.59° (200) aspect became stronger. These strong peaks indicate that the TiN coating mainly grows in the (111) and (200) directions [48], with improved crystalline structure [49]. TiN peaks are assigned following the face-centered cubic phase of TiN [17]. Osbornite phase can be predicted for TiN deposition [49]. The weak peaks in the TiN coating are observed in the (311) plane and the (220) plane, respectively [49]. The (220) orientation of the TiN coating at 61.42° was ignored due to the low diffraction peaks.

Although TiAlN and TiCN coatings have (111) orientations, their density is lower than TiN coatings. However, unlike other coatings, it was observed that there was a significant broadening in the diffraction of the (111) and (200) planes, which are the preferred orientations of the TiN coating [39]. Also, the (111) orientation showed significantly higher intensity than the (200) orientation. The TiN coating deposited by reactive magnetron sputtering has a strongly (111) oriented phase [50]. XRD analysis of TiAlN coating shows that the peaks shift to higher 2θ values. This situation is due to the substitution of Ti atoms in the TiN lattice by Al atoms with smaller atomic radii, thus resulting in shrinkage in the lattice parameter [17]. This study's TiAlN (111) index corresponds to 37.19 degrees. 37.46 degrees [51,52] is specified for a single layer in the literature. Kutschej et al. [51] stated that when aluminum is included in the fcc $Ti_{1-x}Al_xN$ solid solution, smaller Al atoms will replace Ti atoms so that the lattice parameter will shift to low values. Therefore, a significant shift will occur in fcc.

As for the TiCN (JCPDF#42-1488) coatings, according to the XRD results of the TiCN coating, (111), (200), (220) and (311) peaks were observed. Among the corresponding orientations of TiCN, (111) was the strongest and (220) the weakest Bragg plane. XRD results revealed that the TiCN films are polycrystalline materials with FCC lattice [52]. The (220) orientation emerged in all the coatings. In addition, peaks of the Ti6Al4V alloy were

also observed for the coated samples, as the X-ray could penetrate the thin coating layers and reach the substrates [53].

The $\beta$ data (width of the intensity plot) obtained by XRD analysis was used to determine the crystal size of the coated samples. The term crystal size was used when the size of each crystal was <0.1 μm. The average grain size of the coatings was found by applying the Scherrer formula shown in Equation (1) [48,54]. Table 3 presents a comparison of data for all coatings.

$$D = \frac{0.9 \times \lambda}{\beta \times Cos\theta} \tag{1}$$

where the width measure is half the maximum diffraction line length, for the [°2Th] value. The values given in Table 3 were taken as criteria to calculate the value of crystal sizes (*D*).

**Table 3.** Crystallite sizes corresponding to the (111) and (200) planes for the coatings.

| Coatings | Miller Indice | Peak pos. [$^0$2Th] | Fwhm $\beta$ | Crystallite Size D (nm) |
|---|---|---|---|---|
| TiAlN | 111 | 37.19 | 0.314 | 26.6 |
| TiCN | 111 | 36.44 | 0.18 | 46.5 |
| TiCN | 200 | 42.46 | 1.102 | 7.7 |
| TiN | 111 | 36.28 | 1.298 | 6.4 |
| TiN | 200 | 42.59 | 1.143 | 7.4 |

When the crystal sizes of the coatings given in Table 3 are examined, it is seen that the smallest crystal size is in the TiN coating, followed by TiAlN and TiCN. The (111) plane of the TiN coating had the smallest crystal size (in the range of 6.4 nm) compared to the other coatings, as seen in Figure 7. The (200) orientation of the TiN coating also ranked second with an average crystal size of 7.6 nm. For the other coatings, TiCN 46.5 and TiAlN 26.6 nm crystal size was calculated for the (111) plane.

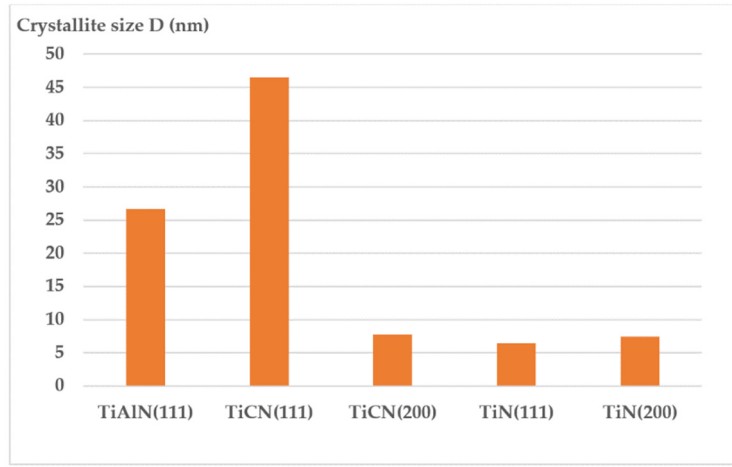

**Figure 7.** The crystallite size of the TiN, TiAlN and TiCN coatings.

Decreased crystal size resulted in a finer dense structure. At the same time, high hardness values were obtained because the dislocation density increased with the decrease in crystal size. The TiN coating showed the highest hardness value (3500 HV) depending on the grain size.

### 3.2. Wear Behavior of the Coatings

In wear tests, wear losses and wear track widths were taken as criteria to determine the wear resistance. In order to measure the wear loss precisely, the samples prepared from Ti6Al4V alloy were cleaned with alcohol before and after the wear test and weighed on a precision balance. The same procedure was applied to the wearing parts (abrasive

disc) used in the wear test. The abrasive disc was changed in each test. Before and after the wear test, the weight losses of each sample and wearing part were measured with 1/10,000 precision with the help of the OHAUS brand precision balance. In pin-on-ring wear tests, experiments were carried out on 24 specimens depending on the load and shear rate. At the end of the wear tests, the average friction coefficients of each sample in the steady-state sliding regime were calculated and the wear track widths were determined by taking SEM images in the wear track region. The sliding distance depends on the speed and time. In wear tests, friction forces (Ft) were recorded using Picolog (Data Logger) depending on the sliding distance. Friction forces were found separately for each sample in the steady-state friction region, as shown in Figure 8, and the friction coefficients were calculated.

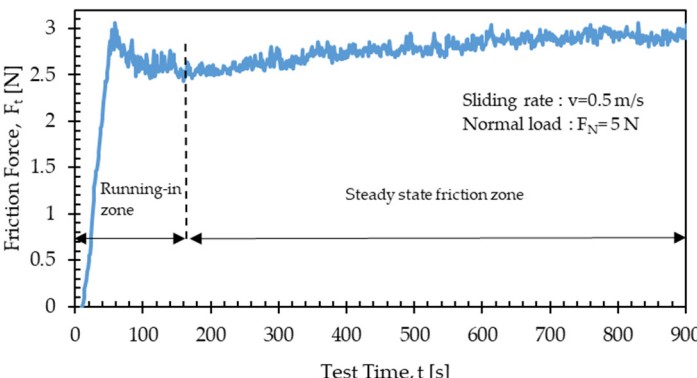

**Figure 8.** The friction force from the steady-state friction zone for TiAlN coating at 5 N normal load and 0.9 m/s sliding rates.

When Figure 8 is examined, the variation in friction force over time for TiAlN coating at 5 N normal load and 0.9 m/s sliding rates are seen. As the first surface is polished, the friction forces increase in the first stage, then slightly decrease and show a steady state. The friction coefficient is calculated using the average value of the friction forces taken in the steady-state regime. It is seen that the TiAlN coating exhibits a long-term process and a high steady-state friction coefficient [15]. The friction coefficient can be divided into two components. One of them is the adhesion component due to the adhesive force and the other is the deformation component due to the plastic deformation force [39].

Equation (2) is used to calculate the coefficient of friction in the wear test. The friction coefficients can be calculated when the average frictional forces obtained in the steady-state region are divided by the normal force, as seen in Equation (2). It is important to determine the time variation in the coefficient of friction during sliding wear in the literature [39].

$$\mu = F_t / F_N \tag{2}$$

where the coefficient of friction ($\mu$) is equal to the ratio of the friction force $F_t$ to the normal load $F_N$. In order to find the friction coefficients, friction forces are continuously recorded by a multi-channel recorder during the experiments.

The results from the wear tests performed on Ti6Al4V alloy and TiN, TiAlN, TiCN coatings are given in Table 4 depending on the load and sliding rates. The table shows the weight losses of the sample and abrasive discs, the wear track widths on the sample surfaces and the friction coefficients.

**Table 4.** Weight loss, wear track widths, friction coefficient values for uncoated Ti alloy and coatings at different loads and sliding rates.

| Materials | Sample No | S. Rate (ms$^{-1}$) | Load [N] | Time (min) | Wear Weight Loss | | Wear Track (mm) | Frictional Coefficient $\mu$ |
|---|---|---|---|---|---|---|---|---|
| | | | | | Pin (mg) | Disc (mg) | | |
| Ti6Al4V | 1 | 0.5 | 5 | 15 | 2.1 | 0.5 | 3.360 | 0.399 |
| | 2 | 0.9 | 5 | 15 | 2.3 | 0.7 | 3.360 | 0.428 |
| | 3 | 0.5 | 15 | 15 | 3.7 | 0.8 | 3.828 | 0.216 |
| | 4 | 0.9 | 15 | 15 | 4.6 | 0.9 | 4.095 | 0.177 |
| | 5 | 0.5 | 30 | 15 | 4.9 | 1.2 | 4.240 | 0.133 |
| | 6 | 0.9 | 30 | 15 | 6.8 | 0.6 | 4.525 | 0.073 |
| TiN | 1 | 0.5 | 5 | 15 | 0.1 | 0.6 | 0.595 | 0.635 |
| | 2 | 0.9 | 5 | 15 | 0.2 | 0.9 | 0.680 | 0.749 |
| | 3 | 0.5 | 15 | 15 | 0.3 | 0.8 | 0.760 | 0.463 |
| | 4 | 0.9 | 15 | 15 | 0.4 | 1.2 | 0.897 | 0.389 |
| | 5 | 0.5 | 30 | 15 | 0.5 | 0.9 | 0.800 | 0.336 |
| | 6 | 0.9 | 30 | 15 | 0.7 | 2.0 | 1.508 | 0.314 |
| TiCN | 1 | 0.5 | 5 | 15 | 0.3 | 0.8 | 0.692 | 0.720 |
| | 2 | 0.9 | 5 | 15 | 0.6 | 1.0 | 1.025 | 0.591 |
| | 3 | 0.5 | 15 | 15 | 0.4 | 0.9 | 2.348 | 0.363 |
| | 4 | 0.9 | 15 | 15 | 0.7 | 0.6 | 2.815 | 0.346 |
| | 5 | 0.5 | 30 | 15 | 2.2 | 0.7 | 3.683 | 0.225 |
| | 6 | 0.9 | 30 | 15 | 8.5 | 1.1 | 4.468 | 0.160 |
| TiAlN | 1 | 0.5 | 5 | 15 | 0.2 | 0.4 | 0.829 | 0.599 |
| | 2 | 0.9 | 5 | 15 | 0.3 | 1.1 | 0.970 | 0.556 |
| | 3 | 0.5 | 15 | 15 | 0.5 | 0.5 | 0.707 | 0.331 |
| | 4 | 0.9 | 15 | 15 | 1.9 | 1.4 | 3.710 | 0.235 |
| | 5 | 0.5 | 30 | 15 | 3.1 | 0.8 | 3.725 | 0.212 |
| | 6 | 0.9 | 30 | 15 | 7.1 | 1.1 | 4.423 | 0.147 |

In order to understand the variation in friction coefficient depending on load and sliding rate, wear tests were conducted under dry friction conditions at 0.5–0.9 m/s sliding rates and a 5–30 N load range. When the friction coefficients in Table 4 are examined, it is seen that the friction coefficient values decrease as the sliding rate and load increase. In the graph shown in Figure 9, the variation in the friction coefficients is compared depending on the sliding rate and normal load.

When the coatings were compared according to their friction coefficient values, it was seen that TiCN and TiAlN coatings had a lower friction coefficient than TiN coatings.

Surface roughness is an important parameter that affects the friction coefficient. In Figure 10a,b, the effect of surface roughness on the friction coefficient is presented. Measurement results showed that TiAlN and TiCN coatings have lower surface roughness values than TiN coatings. Uncoated Ti6Al4V alloy has high roughness. However, the lower coefficient of friction is due to the oxide layer formed on the surface, which is based on passivation. The alloy's SEM image and EDX analysis also reveal this situation, as seen in Figure 11. The coefficient of friction decreases further as the sliding rate increases (Figure 9). Increasing the sliding rate increases the temperature. Increasing friction heat with increasing temperature increases oxidation and the developing oxide layer shows a lubricating effect [15]. When the friction coefficient of the TiN coating was examined, it was found to be higher than the TiCN coating. However, the lower wear rate of the TiN coating indicates that it can perform well at high temperatures [15,55].

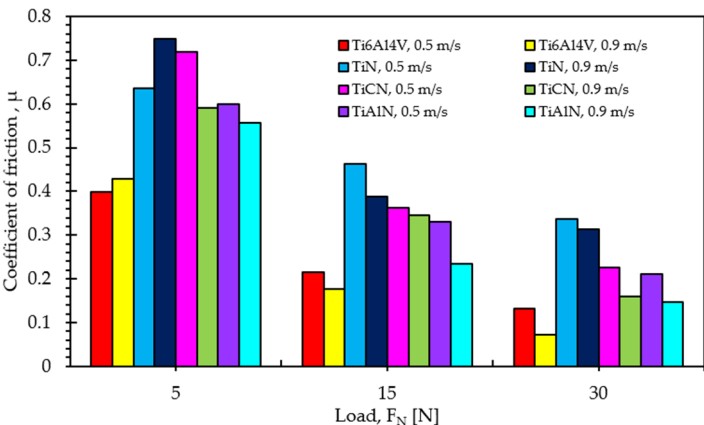

**Figure 9.** The variation in coefficient of friction with load (0.5 m/s–0.9 m/s sliding rate).

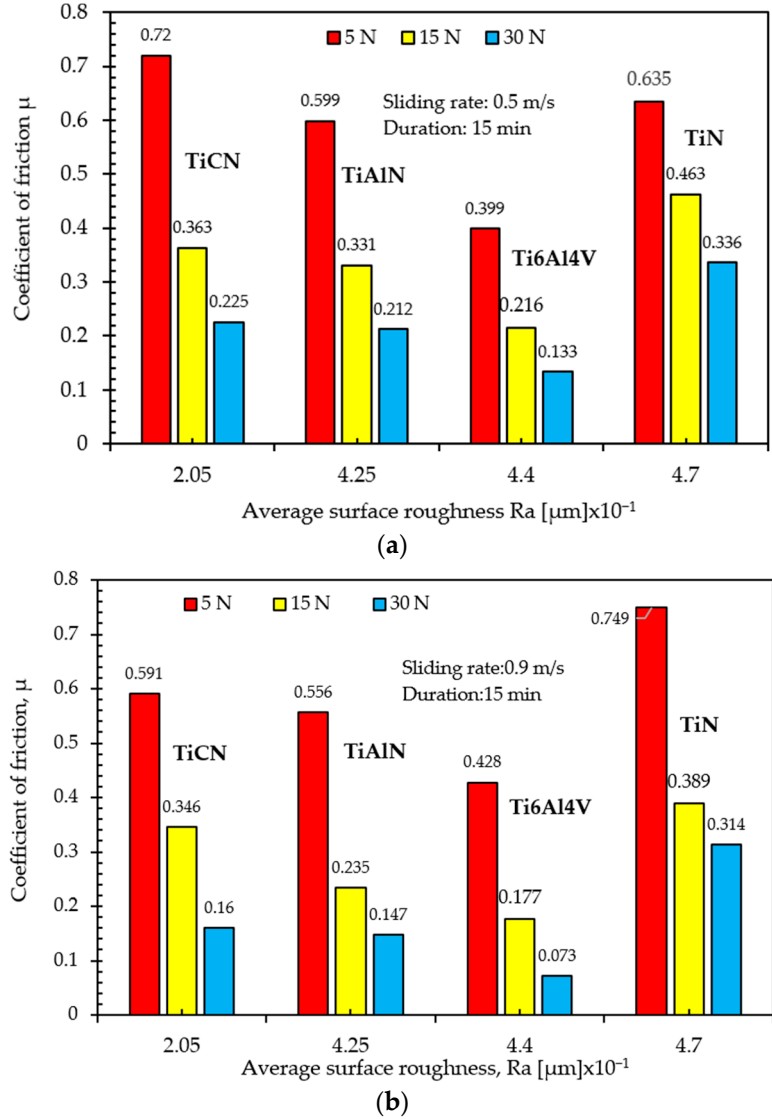

**Figure 10.** The variation in coefficient of friction with surface roughness. (**a**) 0.5 m/s sliding rate, (**b**) 0.9 m/s sliding rate.

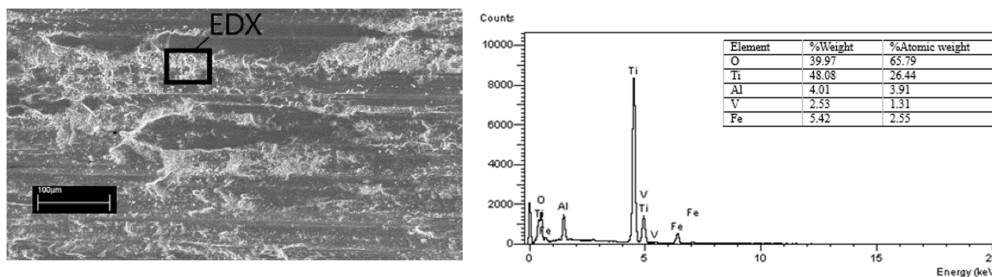

**Figure 11.** Scanning electron microscope image of Ti6Al4V alloy and energy dispersive X-ray analysis results on its surface after wear test.

Wear tests were carried out under the same conditions, using a separate wearing part for each test. Weight loss and wear track widths were used to determine wear resistance. The weight losses were measured for both the samples and abrasive discs using a sensitive scale (1/10,000). SEM microscopy was used to determine the wear track widths as shown in Figure 12. In each experiment, the average values measured in the x and y directions on the wear track were taken as the wear track's width.

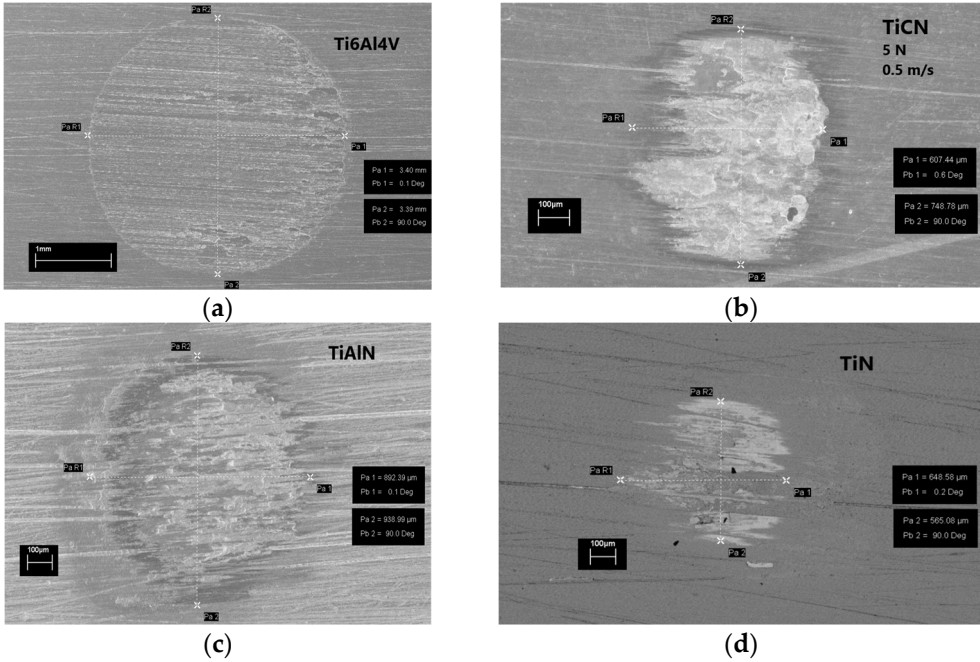

**Figure 12.** Wear track widths determined by scanning electron microscopy at the same load and sliding rate; (**a**) Ti6Al4V, (**b**) TiCN, (**c**) TiAlN, (**d**) TiN.

As a result of the wear test performed with 5 N load and 0.5 m/s sliding rate parameters (Figure 12), TiN coating showed the lowest wear track width, while uncoated Ti6Al4V alloy showed the highest wear track width. Table 4 gives all values of wear track widths for various coated and uncoated samples.

Figure 13 shows the SEM image of the uncoated Ti alloy in the wear zone. In Figure 13a, pore and driving grooves are seen in the wear zone of the uncoated Ti alloy. The wear track width and depth of the Ti6Al4V alloy are significantly higher than the coatings [56,57].

The increase in load and sliding rate increases the wear mass loss, as seen in Figure 13b,c. The increase in sliding rate produces debris, which deforms the surface texture and increases the wear track width. The deformed surface accelerates wear in dry sliding wear, grooving and spallation wear mechanism [56,57].

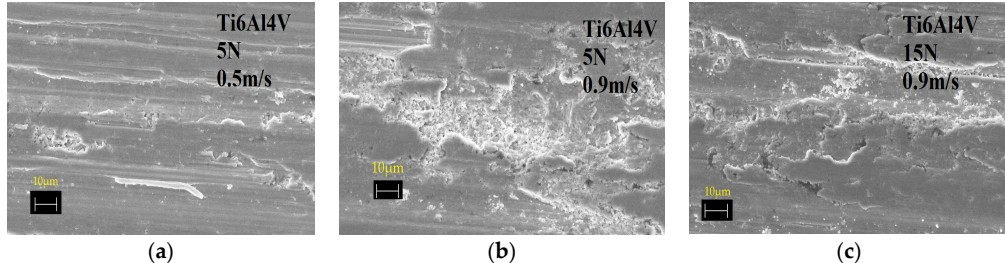

**Figure 13.** Scanning electron microscope image of the uncoated Ti alloy in the wear zone. (**a**) 5 N and 0.5 m/s, (**b**) 5 N and 0.9 m/s, (**c**) 15 N and 0.9 m/s.

After coating treatments, the wear track widths were significantly reduced compared to the uncoated Ti6Al4V alloy [53]. When the wear track width between TiAlN and TiCN coatings was compared, it was found that the wear track width of the TiCN coating was lower than that of the TiAlN coating.

The TiN coating had the lowest wear loss and the lowest wear track widths as shown in Table 4 and Figure 14a,b, respectively.

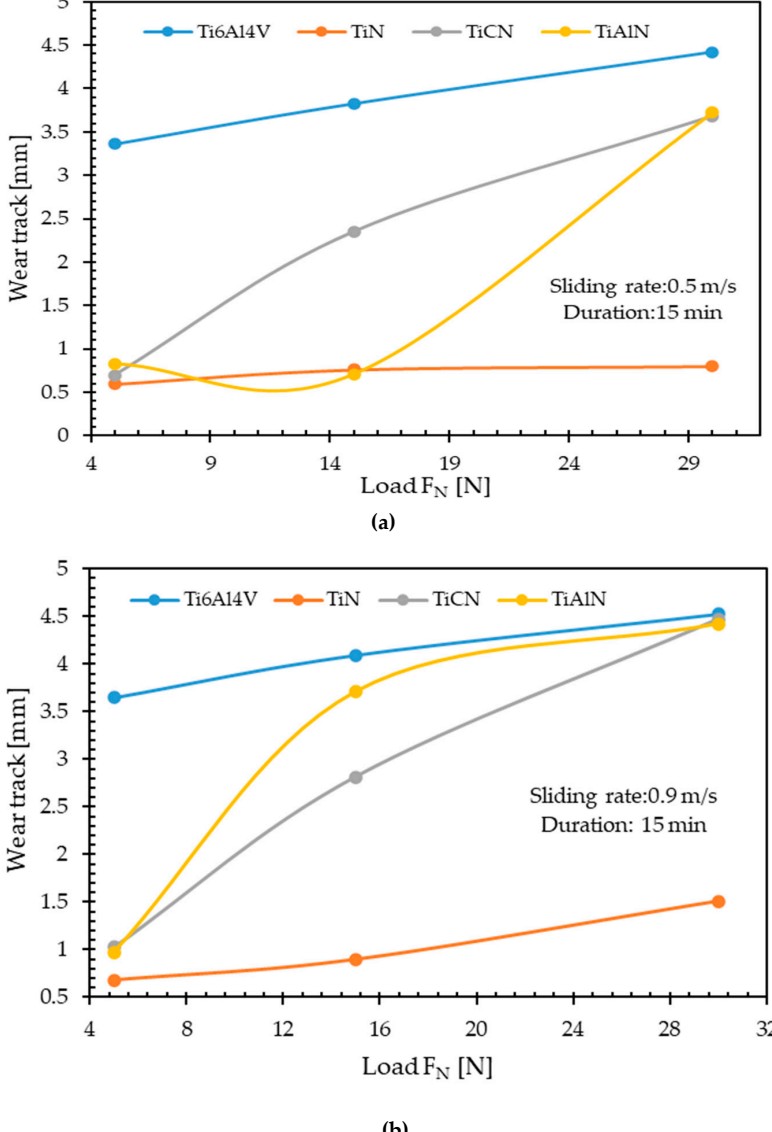

**Figure 14.** The variation in wear track with a load. (**a**) 0.5 m/s sliding rate, (**b**) 0.9 m/s sliding rate.

The wear track width values increased with the increase in load. With the increase in the sliding rate, the wear track width values showed a more considerable increase. Although it is determined that there is a significant difference between uncoated and coated samples at low loads, this difference decreases as the load increases.

The lowest wear rate was achieved in the TiN coating. XRD analyses revealed that fine-grained structures were obtained by coating. The strength of fine-grained structures is higher than coarse-grained ones. TiN showed a small grain size of 6.4 microns in the (111) orientation, so it became the coating with the highest wear resistance. The low wear track widths in TiCN and TiAlN coatings are consistent with the friction coefficient values. In addition, the hardness values of the coatings directly affected the wear resistance [15,54,58].

In Figure 15, the variation in weight losses according to the experimental parameters is given. Graphs are shown with increasing loads and increasing sliding rate. Both parameters result in increase in the weight loss. The highest weight loss is observed in the uncoated samples, as shown in Figure 15.

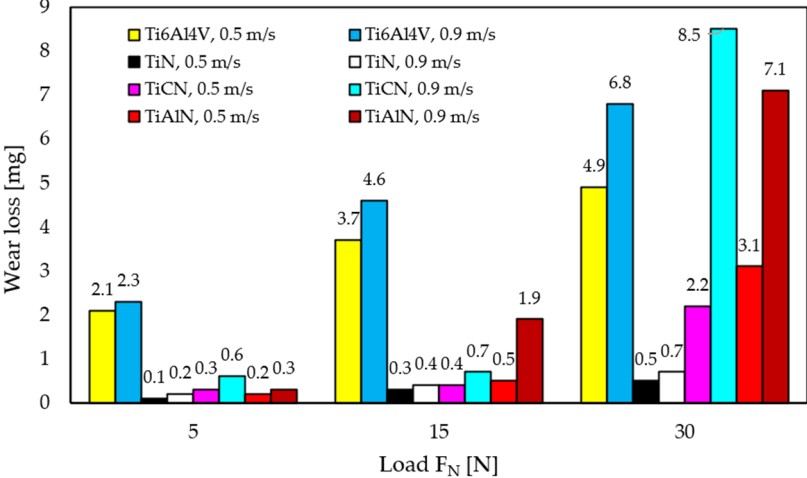

**Figure 15.** The variation in wear loss with load and sliding rate.

As the roughness of the coating decreases and the hardness increases, the adhesion forces decrease and resistance to plastic deformation increases. As seen in Figure 16a, the low hardness uncoated Ti6Al4V alloy exhibited the highest wear loss due to the resultant spall formation [49]. The wear loss increased as the load and sliding rate increased, while the lowest wear loss was found in TiN coating.

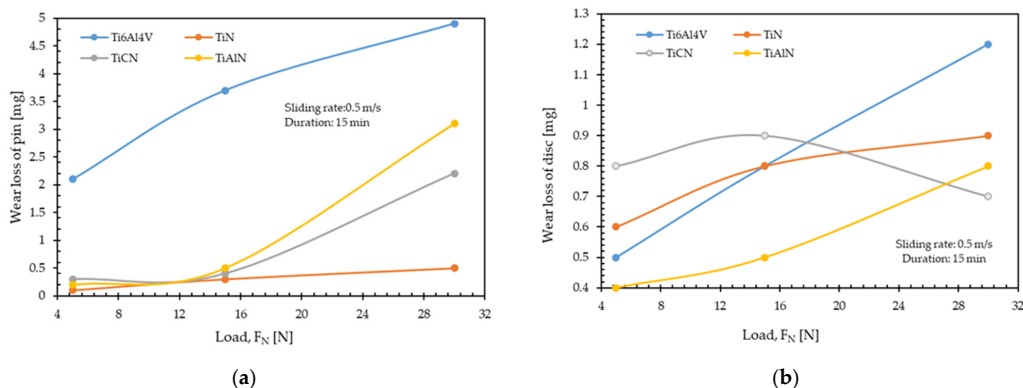

(**a**)  (**b**)

**Figure 16.** (**a**) The variation in wear loss with load for the pin, (**b**) The variation in wear loss with load for the wearing part(disc).

In Figure 16a,b, the wear loss of the abrasive disc used during the wear test is compared with the wear loss of the samples. It is seen that the weight loss of the abrasive disc increases depending on the hardness of the coatings (Figure 16b). The coatings' high hardness and

adhesion strength increase wear resistance [39]. Therefore, the contact state between the sample and the abrasive disc highly depends on the type of coating.

The highest wear loss occurred on the abrasive disc surface during the wear test of the TiN coating. The high-hardness TiN coating eroded the disc surface and formed hard debris, which caused the wear of the three bodies during further sliding [49]. The same mechanism did not affect the disc surface much in TiCN coating wear (Figure 16b). This is because the low hardness and lubricating effect of the TiCN coating reduced the mass loss of the abrasive disc compared to the TiN and TiAlN coatings.

The wear tracks obtained because of the wear tests performed at the same load and sliding rates are shown in Figure 17.

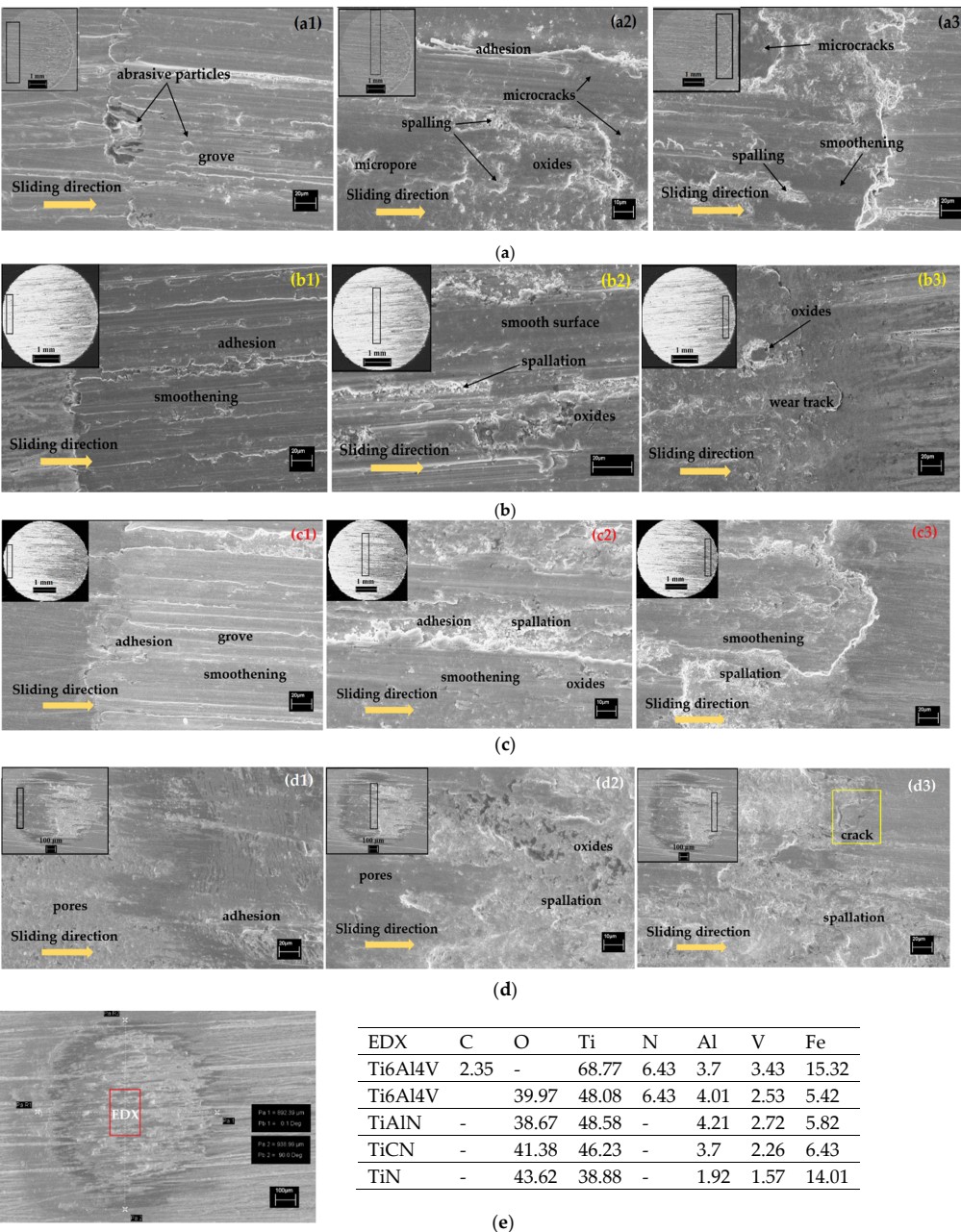

**Figure 17.** The wear track SEM images of (**a**) Uncoated Ti6Al4V, (**b**)TiAlN, (**c**) TiCN, (**d**) TiN, (**a1–d1**) The beginning of wear track, (**a2–d2**) Medium zone, (**a3–d3**) The end of the wear track, (**e**) EDX analysis of the medium zone.

| EDX | C | O | Ti | N | Al | V | Fe |
|---|---|---|---|---|---|---|---|
| Ti6Al4V | 2.35 | - | 68.77 | 6.43 | 3.7 | 3.43 | 15.32 |
| Ti6Al4V | | 39.97 | 48.08 | 6.43 | 4.01 | 2.53 | 5.42 |
| TiAlN | - | 38.67 | 48.58 | - | 4.21 | 2.72 | 5.82 |
| TiCN | - | 41.38 | 46.23 | - | 3.7 | 2.26 | 6.43 |
| TiN | - | 43.62 | 38.88 | - | 1.92 | 1.57 | 14.01 |

The SEM images in Figure 17a–d compare the wear behavior of uncoated Ti6Al4V alloy and coatings. A wear track is in the upper left corner of the SEM images (Figure 17a1–d1). SEM images were taken from three regions examined separately for wear tracks. The first SEM images Figure 17a1–d1 represent the beginning of the wear track, Figure 17a2–d2 represent the middle region of the wear track, and Figure 17a3–d3 represents the end of the wear track. The wear tracks in the left corner of the TiAlN and TiCN coating SEM images are taken from the backscatter images. As given in Table 4, it is seen that the wear track widths decrease significantly depending on the coating type.

Following observations from the wear behavior of uncoated Ti6Al4V alloy were made. During wear, the rotating abrasive disc drags wear residues generated by the local welding effect. Depending on the sliding direction, some of these residues remain at the end of the wear track, while some are carried up to the beginning. Therefore, the wear tracks of the uncoated Ti6Al4V alloy have ridges that accumulate at the beginning and end of the wear track (Figure 17a1–a3), and continuous grooves and scratches occur parallel to the sliding wear direction. Hardness effectively reduces the depth of grooves and scratches [15].

The groove and embedded abrasive particles, which are observed parallel to the sliding direction on the surface, cause a large amount of wear due to the load and sliding rate. Surface microcracks that occur under the influence of shear stresses are formed due to the Ti6Al4V alloy's lower hardness than the abrasive disc. Parallel to the sliding wear direction, plastic deformation zones are observed from place to place. These zones might have formed due to presence of shear stresses parallel to the surface near the exit edge. Moreover, increasing temperatures also trigger the formation of oxides. The wear tracks regions were analyzed with the EDX and presence of oxides was affirmed as measured from the middle region of the tracks (Figure 17e)

Depending on the wear test parameters, the wear track SEM images of the TiAlN coated specimens are shown in Figure 17b1–b3. In the surface analysis of the wear track, it is seen that the wear resistance of the TiAlN coating is better under the same conditions (Figure 17b1–b3). When the middle and end regions of the wear track are examined; it is seen that the wear starts as an adhesive. No crack formation was observed along the track. During wear, oxides were observed in the middle and end regions of the track (Figure 17b2,b3) depending on the direction of sliding because of temperature. The presence of oxides was determined by EDX analysis (Figure 17e) taken from the middle region of the TiAlN coating wear track. A two-body abrasive abrasion effect is seen on the surface, while oxidation of the coatings is due to friction in the wear zone. The wear analyses in the literature show that the wear of Ti-based coatings is caused by the formation of titanium oxide [15,55]. At the same time, small abrasive particles, which are separated from the coating by friction can reveal the sliding wear mechanism [15,54,58].

Compared to the uncoated Ti alloy, smaller wear track widths (Figure 17c1–c3) and thinner grooves were observed in TiCN coatings. While there was general oxidation in the coatings, it was observed that there were no abrasive particles embedded in the wear tracks. The high hardness of the coating reduced the penetration of abrasive particles and increased wear resistance thanks to the tribo layer, which causes a lubricating effect under higher loading conditions [15,49,54,58].

The wear track obtained in TiN coating is smaller than other coatings as shown in Figure 17d1–d3. There are partially glaze layers at the beginning and the end of the coating's wear track. It is understood from the SEM image that the coating could not be removed in the wear zone (Figure 17d1–d3). During dry sliding, the adhesion wear of the TiN coating is reduced, but the oxide particles formed in the next step are broken off, adhere to the disc surface, and cause abrasive wear on the surface. Likewise, cracks in the hard coating layer seen in Figure 17d3 settle on the disc surface and cause three-body abrasive wear. It is clear from the above results that the coatings improve the wear resistance of the Ti6Al4V alloy by increasing the surface hardness. The high hardness of coatings increases abrasive wear [15,55–57]. In general, sliding wear and spalling were observed in all coatings. The

TiN coating was the most effective coating type in reducing the wear rate. Therefore, TiN is the best choice for applications where abrasive wear is the dominant mechanism [15,39,59].

## 4. Conclusions

TiAlN, TiN, TiCN coatings obtained on Ti6Al4V alloy significantly increased the wear resistance as a result of wear tests performed at room temperature in dry conditions.

When the surface roughness and friction coefficient values affecting the wear resistance were examined, TiN > Ti6Al4V > TiAlN > TiCN and TiN > TiCN > TiAlN > Ti6Al4V were determined, respectively. According to the average steady-state friction coefficients, the coatings increased the friction coefficient due to their increased roughness, but the Ti6Al4V alloy had a lower friction coefficient thanks to its passivation ability and the oxide layer it formed on the surface.

The coatings significantly increased the hardness of the Ti6Al4V alloy, resulting in the order of TiN > TiAlN > TiCN > Ti6Al4V according to the coating type. The TiN coating with the smallest crystal size increased the hardness of the alloy six times and improved its wear resistance.

Considering the wear track widths and wear losses at different loads and sliding rates, the lowest wear track width was observed in TiN coating, and the wear track widths increased with increasing load and sliding rate. The type of coating greatly influenced the contact state between the abrasive disc and the coating. Depending on the surface properties of the coatings, adhesion, abrasion and oxidation wear mechanisms were observed. However, since triple abrasive wear was found to be more effective, it was revealed that TiN coating has the potential to meet expectations in applications where abrasive wear is dominant.

**Author Contributions:** Conceptualization, Methodology, Formal analysis, Ş.D. and M.T.; Investigation, Ş.D. and M.T.; Writing—original drafting, Ş.D.; Review, images editing, graphic editing; D.O. and M.T.; Formal analysis of coatings, Ş.D. and M.T.; Project management, Ş.D.; Funding acquisition, Ş.D. and M.T.; Resources, Ş.D. All authors have read and agreed to the published version of the manuscript.

**Funding:** This research was funded by Erciyes University Scientific Research Units (Project No: FBY-09-852) and is gratefully accepted.

**Institutional Review Board Statement:** Not applicable.

**Informed Consent Statement:** Not applicable.

**Data Availability Statement:** Data sharing not applicable.

**Acknowledgments:** The authors thank Erciyes University Scientific Research Unit for supporting the research and Erciyes University Technology Research and Application Center for microstructure analysis. Many thanks to Mechanical Engineer Hamza Taha Aydemir, who made the technical drawings to create the schematic images of the magnetron sputtering method.

**Conflicts of Interest:** The authors declare no conflict of interest.

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
