# Peer review of "The Effect of TiN, TiAlN, TiCN Thin Films Obtained by Reactive Magnetron Sputtering Method on the Wear Behavior of Ti6Al4V Alloy: A Comparative Study"

_coatings, doi:10.3390/coatings12091238_

Round 1

Reviewer 1 Report

Could not understand what is the use of this statement in the abstract(NOT clear either)"Ti6Al4V alloy must be reliable and long-lasting in industry and the human body."

Avoid abbreviations in the abstract, table/figure captions, and conclusions; hard to follow as readers first glance these.

Define any abbreviations before their first use.

Clear and concise jargon should be used.

The conclusions section should only have novel findings, not the procedures; "TiAlN, TiN, TiCN coatings were coated on Ti6Al4V alloy by the unbalanced reactive magnetron sputtering method and their wear resistance was compared in dry conditions at room temperature." This is part of the methods section. better to have bullet point conclusions; avoid redundant wordings.

Avoid colloquial jargon.

Authors' previous publication; https://iopscience.iop.org/article/10.1088/1757-899X/295/1/012044

has some connection and that needs to be properly cited and not copied statements/paragraphs from that.

Author Response

Dear Reviewer,

In line with the suggestions, the study was reconsidered 

Thank you for your helpful suggestions to improve the study.

Reviewer 2 Report

The manuscript carried out very detailed and interesting comparative study on the wear behavior of TiN, TiAlN, TiCN thin PVD films and its substrate alloy Ti6Al4V, it showed that the TiN film present optimum wear characters for its high hardness from coating density and fine crystal microstructures. Despite the practical importance of the main problem of the paper – the optimum wear performance and mechanisms, it shall be minor revised to meet the detail requirement as below:

 1. In the introduction part: the authors literature the Ti6Al4V alloy usage on its compatibility with biological tissues, however, the wear behavior was investigated in dry conditions, it not make sense of the motivations of the study.

2. In material and methods part: TiCN coating was carried out in the order of Ti interlayer +TiN coating + TiCN coatings, what is the process on TiAlN?

3. In the experimental result and discussion part: (Lines 181 – 185), why the authors present Plasma nitriding? The hard TiN film is from the PVD TiN film that shall be deposited on the Ti interlayer, not from the plasma nitriding. It is confused, at least not clear.

4. In Fig. 9 and Fig. 10 , shall we distinguish the different performance histograms of the coatings by line types, rather than just color?

5. Line 494, …due to the alloy’s lower hardness than the abrasive disc, shall the Ti6Al4V alloy’s lower hardness….

Author Response

(The authors gave the same response as above.)

Reviewer 3 Report

The manuscript “The Effect of TiN, TiAlN, TiCN Thin Films Obtained by Reactive Magnetron Sputtering Method on the Wear Behavior of 3 Ti6Al4V Alloy: A Comparative Study” presents many experimental data that are explained and interpreted in an original and  interesting manner, so that they can be landmarks of real interest for specialists in the field. Unfortunately, the manuscript  need a serious drafting review to reach the needed level of a scientific paper that can be published in a high-impact journal such is "Coatings". Because there are too many misspelled phrases, I have underlined them so that the authors to try to modify them according to the criteria of the logical (& grammatical) basic exposition of the results in a research paper.

Author Response

Dear Reviewer,

In line with the suggestions, the study was reconsidered and edited.

Thank you for your helpful suggestions, which allowed to significantly improve the work.

The underlined lines have been rearranged and shown as painted on the manuscript.

Round 2

Reviewer 1 Report

Writing style should be improved as well as scientifc soundness in reporting.

e.g., Hardness test results should be reported as 384HVxx - response to stress by the material may be different with the stress/force applied or time duration for the indentation etc. Therefore it should be clearly reported as such. 

Too general and long conclusion; they should be derived from this study, and in clear and concise form.

Author Response

In line with the suggestions, the study was revised and edited. The editing certificate is attached.

Thank you for your helpful suggestions to improve the study.

Reviewer 3 Report

The authors have made substantial changes so that I think the article is much better than the first version and can be published in this form.

Author Response

(The authors gave the same response as above.)

Round 3

Reviewer 1 Report

Decimal separator should be correctly used instead of a comma. e.g., 5.33 not 5,33. Please correct all of the figures. Quality of the images should be improved.

Conclusion does not reflect an outcome of a good scientific report; should be novel.

Authors are encouraged to read a few highly cited publication and how to write a good scientific paper. 

Authors have considered the previous comments but still careless mistakes and therefore the credibility of the report is questioned! Please read and check the whole manuscript again world by word and check all of the figures and/or tables.

Reviewers comments should be applied to the whole manuscript not just for what they have mentioned as an example.

The manuscript still has some useful/great content but needs to be modified.
